# Measuring Myotonia: Normative Values and Comparison with Myotonic Dystrophy Type 1

**DOI:** 10.3390/neurolint17080118

**Published:** 2025-07-31

**Authors:** Andrea Sipos, Milán Árvai, Dávid Varga, Brigitta Ruszin-Perecz, József Janszky, Nándor Hajdú, Endre Pál

**Affiliations:** 1Department of Neurology, Medical School, University of Pécs, 7623 Pécs, Hungary; sipos.andrea@aok.pte.hu (A.S.); arvaimilan11@gmail.com (M.Á.); varga.david@pte.hu (D.V.); brigittaperecz@gmail.com (B.R.-P.); janszky.jozsefd@pte.hu (J.J.); 2Institute of Psychology, Eötvös Loránd University, 1053 Budapest, Hungary; hajdu.nandor93@gmail.com

**Keywords:** myotonic dystrophy, myotonia, dynamometry, nine-hole peg test, normative values

## Abstract

Introduction: Myotonia is a rare neuromuscular condition characterized by impaired muscle relaxation. In this study, we provide normative values for clinical tests related to myotonia in the Hungarian population and compare them to patients with myotonic dystrophy type 1 (DM1). Methods: Relaxation tests (10 eye openings, tongue extension, and palm openings), handgrip strength, and the nine-hole peg test were conducted on 139 healthy individuals and 31 patients with DM1. Results: We observed non-significant declines in handgrip strength and relaxation tests with age (*p* < 0.05). Significant differences were found between controls (n:139) and patients with DM1 (n = 31) in all tests (*p* < 0.05). Sex differences were noted in the healthy population: men (n:68/139) had stronger handgrip (mean of men 42.45 ± 1.15 vs. women 24.3 ± 0.9) and slower relaxation tests (mean of eye openings in men 3.6 ± 0.2 vs. in women 4.2 ± 0.2, tongue extensions in men 3.7 ± 0.2 vs. in women 4.2 ± 0.2, palm openings in men 4 ± 0.2 vs. in women 4.9 ± 0.2 However, these differences were not present among patients with DM1. Discussion: Normal values for relaxation tests across different age groups were established. These results might be useful for further clinical investigations. Our study supports the usage of averages of healthy population instead of age groups of relaxation tests and their clinical relevance in the evaluation of patients with myotonia.

## 1. Introduction

Myotonic dystrophy type 1 (DM1) is the most common adult-onset muscular dystrophy, with a prevalence ranging from 0.5 to 18.1 per 100,000 individuals [1]. It is inherited in an autosomal-dominant manner and is caused by the accumulation of trinucleotide repeats at the non-translated end of chromosome 19. In unaffected individuals, the CTG trinucleotide repeat typically comprises fewer than 37 copies. In contrast, individuals with myotonic dystrophy type 1 (DM1) generally exhibit expansions exceeding 50 repeats; the most severe phenotypes are associated with expansions reaching several thousand repeats. There is a loose correlation between repeat length and both age of onset and disease severity, reflecting a phenomenon known as anticipation, wherein clinical manifestations appear earlier and with increased severity in successive generations. The expanded CUG repeats within the mutant *DMPK* mRNA form stable hairpin structures that aberrantly sequester RNA-binding proteins. This sequestration disrupts the regulation of alternative splicing across a wide range of target pre-mRNAs, leading to the misexpression of functionally critical isoforms in muscle and other tissues. The resulting molecular dysregulation contributes to the complex, multisystemic phenotype of DM1, which includes muscle wasting and weakness progressing from distal to proximal, delayed muscle relaxation (myotonia) and systemic symptoms, such as early-onset cataracts, cardiac arrhythmias, insulin resistance, type 2 diabetes mellitus (T2DM), and frontal balding [2,3,4].

DM1 is marked by a gradual progression, with various assessments available to map its course. Muscle strength can be evaluated through Manual Muscle Testing (MMT), though this method is prone to examiner bias and is considered less objective than dynamometry. MMT employs the Medical Research Council (MRC) scale to assess muscle groups based on function (e.g., flexors/extensors) and anatomical location (e.g., neck flexors). To enhance sensitivity and simplify data management, the traditional 0–5 MRC scale was expanded with plus/minus modifiers and then converted to a 1–10 scale. Dynamometry is an instrumental test of muscle strength that provides objective muscle strength measurement using calibrated instruments like the microFET2 (Hoggan Scientific) for limb muscle strength assessment and a Kern dynamometer for grip strength evaluation. Results are reported in Newtons or kilograms. Myotonia may manifest in the hand muscles as grip myotonia or percussion myotonia, which occurs after muscle tapping with a reflex hammer. To assess myotonia severity, relaxation tests can be performed on the most-affected muscles, such as those of the eye, hand, and tongue. Relaxation times are further quantified through timed functional tasks, including repetitive eye opening, tongue protrusion, hand opening, and sit-to-stand maneuvers, providing an indirect measure of myotonia severity. The “warm-up phenomenon,” characterized by reduced relaxation time with repeated contractions, is often observed. Patients with DM1 face significant activity limitations; thus, their disability should also be evaluated using functional tests like the nine-hole peg test (NHPT) and the six-minute walk test (6MWT). Fine motor function and coordination are assessed using the nine-hole peg test (NHPT), which requires patients to insert and remove pegs from a board as quickly as possible, with performance times recorded in seconds. Ambulatory function is evaluated using the six-minute walk test (6MWT), which serves as a comprehensive measure of lower limb strength and endurance. The total distance walked in six minutes is recorded, and results are compared to normative data from healthy Caucasian adults, where average distances of approximately 659 ± 62 m have been reported. Several physiological and anthropometric variables, including height, sex, and forced expiratory volume (FEV1), have been found to influence 6MWT performance, highlighting the need to contextualize findings based on individual characteristics [5].

The impact of the disease on daily life can be assessed using questionnaires like DM1-activ and the 36-Item Short Form Survey (SF-36). The DM1-Activ and DM1-Disability scales are utilized to evaluate disease activity and disability. The DM1-Activ scale measures activity limitations across 20 tasks of increasing difficulty, assigning scores based on the effort required or inability to perform the task. A maximum score of 40 points can be achieved, with lower scores indicating greater functional impairment. The DM1-Disability scale consists of 21 items divided into domains, including neuropsychology, motor function, and daily living activities. The total possible score is 82, with higher scores reflecting more severe disability. Notably, only three of the four questions within the myotonia domain contribute to the total score. Health-related quality of life is assessed using the 36-Item Short Form Survey (SF-36), which examines multiple health dimensions such as physical functioning, role limitations, emotional well-being, fatigue, and general health perception. Each domain is scored on a 0–100 scale, with higher scores indicating better self-reported health status. Data can be analyzed using an online scoring platform. Depressive symptoms, commonly observed in patients with DM1, are assessed using the Beck Depression Inventory (BDI). This instrument comprises 13 items, each with three response options, allowing for a total score ranging from 0 to 39. Higher scores indicate more severe depressive symptomatology. Cognitive evaluations include the Montreal Cognitive Assessment (MoCA) and the Addenbrooke Cognitive Examination (ACE). The MoCA is a brief screening tool designed to detect mild cognitive impairment, covering domains such as executive function, attention, memory, and language. A score of ≥26 out of 30 is considered within normal limits. The ACE provides a more comprehensive evaluation across five domains (attention, memory, verbal fluency, language, and visuospatial skills), yielding a total score of 100, with higher scores indicating superior cognitive performance. Laboratory tests include the analysis of serum creatine kinase (CK), which in DM1 is typically within the normal range or only mildly elevated, unlike in other myopathies. Endocrine and metabolic evaluations are crucial for differential diagnosis and assessment of systemic involvement. Thyroid function tests (TSH, fT4, and fT3) are conducted to exclude thyroid-related myopathies. Glucose metabolism is assessed through fasting glucose and insulin levels, with insulin resistance calculated using the HOMA-IR formula. A value exceeding four indicates significant insulin resistance, a common finding in patients with DM1. Lipid profiles (total cholesterol, HDL, LDL, and triglycerides) are also measured due to the frequent dyslipidemia observed in patients with DM1.

In summary, the comprehensive evaluation protocol employed in the follow-up of patients with DM1 integrates both quantitative and qualitative assessments across neuromuscular, cognitive, psychiatric, and systemic domains. This multidimensional approach not only allows for the precise characterization of disease burden and progression but also informs clinical decision-making and the evaluation of therapeutic interventions.

The aim of our study is to determine normal values for relaxation tests and to assess the influencing factors (age and sex). We compared the results of our patients with DM1 to those of healthy subjects to evaluate the influence of disease-specific variations [6].

## 2. Patients and Methods

Our study was approved by the regional ethical committee (ID number: 53593-5/2020/EÜIG). Participation in our study was voluntary. The healthy group was selected based on medical history, excluding individuals with neurological diseases or other conditions affecting the musculoskeletal system or reducing exercise capacity. For our patients with DM1, the inclusion criterion was a positive genetic test. All patients had one allele with a normal number of CTG repeats and another with a CTG repeat expansion greater than 37, but the exact number of expanded repeats was not determined. Exclusion criteria included severe cardiopulmonary and musculoskeletal impairments that interfere with standard test performance, pregnancy, and residual symptoms from other neurological diseases that might affect the results.

Test recordings were performed by the same investigators (AS, ÁM, BP) according to a standard protocol as part of the annual follow-up. Controls and patients were investigated in the morning, ensuring they had not engaged in significant physical activity beforehand.

### 2.1. Grip Strength

Since DM1 predominantly affects the distal muscles, with pronounced muscle wasting and weakness of hand and finger flexors as well as foot dorsiflexors, we paid special attention to these assessments. Handgrip dynamometer (Kern und Sohn) was calibrated and used as follows: the elbow was bent at 90 degrees, in a middle position, with the forearm held and the wrist also in a middle position. The subject was asked to squeeze the device until they heard a beep, after which they could release it [7]. Both the dominant (DH) and non-dominant hand (NDH) were measured three times, and the results were recorded in kilograms, with the best result considered [8,9,10,11,12,13].

### 2.2. Relaxation Tests

We investigated myotonia in the most-affected muscle groups: eye muscles, masticatory muscles, tongue muscles, and hand muscles. Subjects were asked to perform 10 muscle contractions and relaxations as quickly as possible from the given starting position (eyes closed, mouth closed, palms closed), with results given in seconds. Patients performed these tests only once to eliminate the “warm up” phenomenon. Healthy subjects were allowed to perform these tests three times to reduce the learning effect, with the best result considered [14].

### 2.3. Nine-Hole Peg Test (NHPT)

The nine-hole peg test (NHPT) was used to assess the condition of hand muscles and dexterity. There are a number of factors to be considered when performing and interpreting this examination, which are also components of dexterity. Beyond sensory and motor functions, the presence of myotonia can be considered as an influencing factor as well. The test was performed with both hands, and results are given in seconds. Patients performed the task only once to eliminate the “warm up” phenomenon effect, while healthy subjects, being unfamiliar with the task, were allowed to perform it three times to reduce the learning effect, with the best result considered [10,15,16,17,18]. To measure the rate of impairment of our patients, the Muscle Impairment Rating Scale (MIRS) was recorded for each of them.

### 2.4. Statistical Analysis

In order to measure the effect of DM1, sex, and age on grip strength, eye openings, palm openings, sticking one’s tongue out, the nine-hole peg test, multiple regression analyses were conducted. In the case of the MIRS scores, the independent variables were sex and age. We conducted the statistical analyses using R version 4.3.3 (R Core Team, 2024). The significance level was set at α = 0.05.

## 3. Results

Our study included 139 healthy subjects and 31 patients with DM1. The basic characteristics of participants are shown in Table 1.

The recorded values of dynamometry and functional tests are presented in Table 2.

Results by age group are shown in Appendix A. Using the fitted regression models, we generated predicted motor test scores for hypothetical individuals across all combinations of the predictor variables (e.g., 20-year-old healthy man, 20-year-old woman with DM1, 30-year-old man with DM1). These predictions were compiled into a comprehensive reference table to serve as a clinical decision-support tool, enabling clinicians to rapidly assess patient performance relative to age- and sex-matched healthy controls and to establish norms for comparable DM1 patients.

The results of the regression analyses, including regression coefficients, F-tests, and determination coefficients, are presented in Table 3. It also includes the results of Shapiro–Wilk normality tests for each dependent variable across all study groups. These results support the appropriateness of our modeling approach and provide transparency regarding the distributional assumptions underlying our analyses. Patients with DM1 demonstrated lower grip strength in both their dominant and non-dominant hands compared to healthy individuals. Additionally, grip strength was lower in women than in men, and it decreased with age. Figure 1 contains visualizations of grip strength for both the dominant and non-dominant hands, categorized by DM1 and healthy groups, as well as by sex.

Significant differences were observed in relaxation tests between the healthy group and the DM1 group, with patients with DM1 taking more time to perform the relaxation tasks ten times. Women were significantly slower than men in the eye-opening and palm-opening tasks, although no sex differences were found in the tongue-sticking task. Age had a minimal effect on the performance of these tasks, with older individuals completing them slightly more slowly.

Figure 2 contains relaxation test results categorized by DM1 and healthy groups as well as by sex.

As the results of the nine-hole peg test showed a log-normal distribution, we used the base 2 logarithm of the values in our models. Patients with DM1 took longer to complete the NHPT with both their dominant and non-dominant hands. Sex also had a significant impact on completion time, with women taking longer than men. Age had a significant but small effect on NHPT completion times (Figure 3).

Regarding the MIRS scores of patients with DM1, both age and sex are significant predictors (Figure 4).

We investigated the relationship between recorded values and MIRS. As expected, close correlation was found between NHPT and handgrip strength, as well as with MIRS, as shown in Table 3.

## 4. Discussion

This is the first study reporting normative values of relaxation tests on a healthy population using standardized protocols [19,20].

The present study provides comprehensive evidence for significant motor impairments in patients with myotonic dystrophy type 1 (DM1) across multiple functional domains, confirming and extending previous research on the motor phenotype of this multisystem disorder. Our findings demonstrate that patients with DM1 exhibit markedly reduced grip strength, prolonged muscle relaxation times, and impaired fine motor coordination compared to healthy controls, with these deficits showing characteristic patterns related to age and sex.

### 4.1. Grip Strength Impairments

The substantial reduction in grip strength observed in patients with DM1 (approximately 24.6 kg and 22.5 kg decreases for dominant and non-dominant hands, respectively) aligns with the well-established understanding of DM1 as a progressive myopathy primarily affecting distal muscles. These findings are consistent with previous studies demonstrating that grip strength serves as a reliable and sensitive marker of disease severity in patients with DM1 [21]. The bilateral nature of the impairment, with relatively similar deficits in both dominant and non-dominant hands, reflects the symmetric muscle involvement characteristic of DM1, distinguishing it from other neuromuscular conditions that may show more asymmetric presentations.

The strong relationship between grip strength and age observed in both patient and control groups (coefficients of −0.287 and −0.259 kg per year) demonstrates that patients with DM1 experience the dual burden of disease-related weakness superimposed on normal age-related muscle strength decline. This has important implications for longitudinal monitoring, as clinicians must distinguish between expected age-related changes and disease progression. The sex differences observed, with women showing approximately 14 kg lower grip strength than men, are consistent with established normative data and likely reflect both physiological differences in muscle mass and the complex interaction between CTG repeat length, hormonal factors, and phenotypic expression in DM1.

### 4.2. Myotonia and Muscle Relaxation

The prolonged relaxation times observed in patients with DM1 across all three tested modalities (eye opening, palm opening, and tongue protrusion) provide objective quantification of myotonia, one of the pathognomonic features of DM1. The increases of 4.5, 6.0, and 4.5 s for these respective tasks represent substantial functional impairments that likely translate to significant daily life difficulties. These findings emphasize that myotonia assessment should extend beyond traditional percussion myotonia to include functional measures that better reflect patient experiences.

The relatively modest age effects on relaxation times suggest that myotonia may plateau or progress more slowly than strength-related parameters in DM1, which contrasts the more pronounced age-related changes observed in grip strength. This differential progression pattern may reflect distinct underlying pathophysiological mechanisms, with myotonia primarily resulting from altered chloride channel function due to DMPK gene effects, while weakness stems from progressive muscle fiber degeneration and replacement.

The sex differences in relaxation times, particularly for eye- and palm-opening tasks, represent a novel finding that warrants further investigation. These differences may reflect hormonal influences on membrane excitability or differences in task-specific muscle group involvement between men and women.

### 4.3. Fine Motor Coordination

The nine-hole peg test results provide compelling evidence for fine motor impairment in patients with DM1, with completion times nearly 10 and 9 s longer for dominant and non-dominant hands, respectively. This represents a substantial functional deficit, as the NHPT is specifically designed to assess the complex integration of strength, dexterity, and coordination required for activities of daily living. The correlation between NHPT performance, grip strength, and MIRS scores validates the interconnected nature of motor impairments in DM1 and supports the concept that multiple motor domains are affected by the underlying pathophysiology. This multi-domain impairment pattern distinguishes DM1 from conditions affecting primarily strength or coordination in isolation.

### 4.4. Clinical Assessment Integration

The development of predictive models incorporating age, sex, and disease status represents a significant advance toward precision medicine approaches in DM1 management. The high R^2^ values (0.734–0.743 for grip strength models) indicate that these readily assessable variables account for the majority of variance in motor performance, supporting their utility as clinical predictors. The reference table generated from these models provides clinicians with evidence-based expectations for patient performance, facilitating more accurate assessment of disease severity and progression.

### 4.5. Inter-Measure Correlations and Clinical Validation

The correlation analysis revealed meaningful relationships between motor assessments that provide important insights into the interconnected nature of DM1 motor impairments. The strong correlation between dominant and non-dominant grip strength (r = 0.95) confirms the bilateral and symmetric nature of muscle weakness in DM1, supporting the validity of using either measure as a representative indicator of overall strength capacity. Similarly, the moderate correlation between nine-hole peg test performance for both hands (r = 0.78) suggests that while there may be some hand-specific variations, fine motor coordination deficits generally affect both sides comparably.

The negative correlations between grip strength and nine-hole peg test completion times (r = −0.47 to −0.56) demonstrate the expected relationship, where stronger grip strength is associated with faster task completion. This relationship validates both measures as complementary assessments of motor function, with grip strength reflecting raw force generation capacity and NHPT capturing the complex integration of strength, dexterity, and coordination required for functional tasks.

Most importantly, the correlations with MIRS scores provide strong evidence for the clinical validity of these objective motor measures. The strong negative correlations between MIRS and grip strength (r = −0.64 to −0.68) indicate that higher disease severity ratings correspond to greater strength impairments. Similarly, the positive correlations between MIRS and nine-hole peg test times (r = 0.31 to 0.48) show that patients with higher clinical impairment ratings require more time to complete fine motor tasks. These relationships validate the use of objective motor measures as quantitative indicators of disease severity that complement traditional clinical rating scales.

The strength of these correlations supports the concept that while each assessment captures distinct aspects of motor function, they collectively reflect a common underlying pathophysiology. This interconnected relationship pattern suggests that comprehensive motor assessment batteries may be more informative than individual measures alone and supports the potential for developing composite motor outcome measures that could improve sensitivity for detecting disease progression in clinical trials and routine care.

### 4.6. Pathophysiological Implications

The comprehensive motor impairment pattern observed in our DM1 cohort reflects the multisystem nature of DMPK gene expansion effects. The combination of myotonia (reflecting altered ion channel function), weakness (reflecting muscle fiber degeneration), and coordination deficits (potentially reflecting both peripheral muscle effects and central nervous system involvement) illustrates the complex pathophysiology underlying DM1. The relatively preserved statistical power across different outcome measures suggests that each assessment captures distinct but related aspects of disease impact.

### 4.7. Clinical Applications and Future Directions

These findings have immediate clinical applications for DM1 assessment and monitoring. The quantitative nature of all measures supports their use as objective outcome measures in clinical trials, while the development of normative reference data enables more precise clinical assessment. The sex and age corrections provided by our models allow for more accurate interpretation of individual patient results.

Future research should investigate the longitudinal progression patterns of these motor measures to establish their utility as progression biomarkers. Additionally, the relationship between these objective motor measures and patient-reported outcomes deserves investigation to ensure clinical relevance of assessment batteries.

## 5. Limitations

Several limitations should be acknowledged. The cross-sectional design limits our ability to assess progression patterns, and the relatively small sample size for MIRS analysis (n = 31) may limit the generalizability of these specific findings. The absence of precise CTG expansion data in the DM1 group represents a significant limitation, as repeat length is known to correlate with disease severity and phenotypic expression and would have enabled investigation of important genotype–phenotype relationships.

The use of a convenience sample from a single center may affect the generalizability of our findings to broader DM1 populations, as patient characteristics, disease management approaches, and demographic factors may vary across different clinical settings and geographic regions. Additionally, possible learning effects in the control group cannot be entirely excluded, despite attempts to minimize them through standardized instructions and practice trials. Such effects could influence the magnitude of the observed differences between groups.

Furthermore, the lack of test–retest reliability assessment for the clinical tools used in this study limits our understanding of measurement precision and the ability to distinguish true clinical change from measurement error in future longitudinal applications. Establishing reliability metrics would be essential for implementing these measures as progression biomarkers in clinical trials or routine monitoring.

## 6. Conclusions

This study provides robust evidence for comprehensive motor impairment in patients with DM1 across strength, myotonia, and coordination domains. The development of predictive models and reference standards represents a significant advance toward standardized, objective assessment in DM1 clinical care and research. These findings support the use of multi-domain motor assessment batteries in DM1 and provide the foundation for future longitudinal studies of disease progression.

## Figures and Tables

**Figure 1 neurolint-17-00118-f001:**
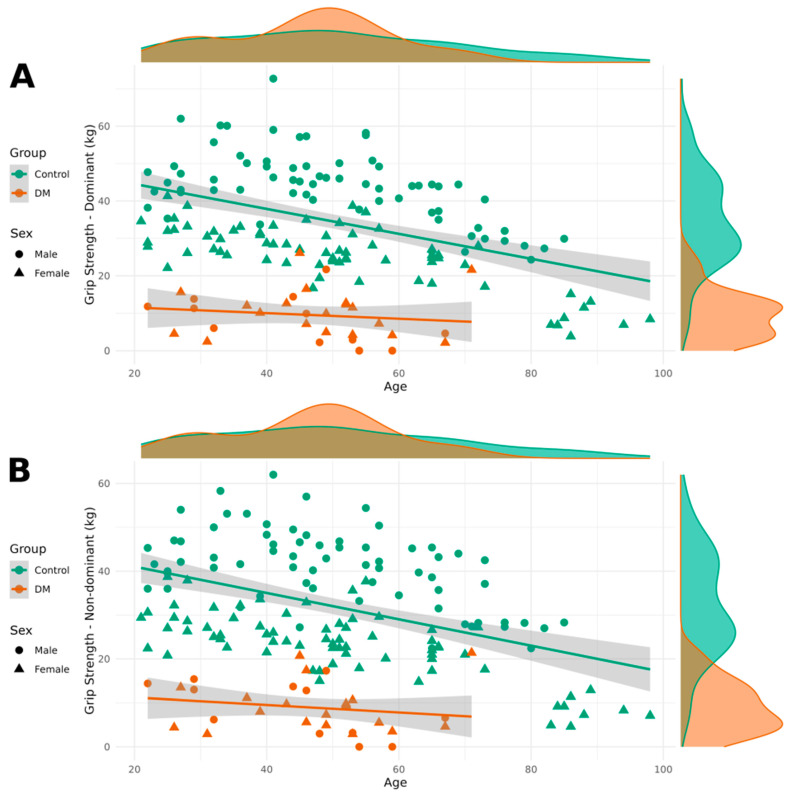
Grip strength of the dominant hand (**A**) and the non-dominant hand (**B**) categorized by patient and control groups as well as by sex.

**Figure 2 neurolint-17-00118-f002:**
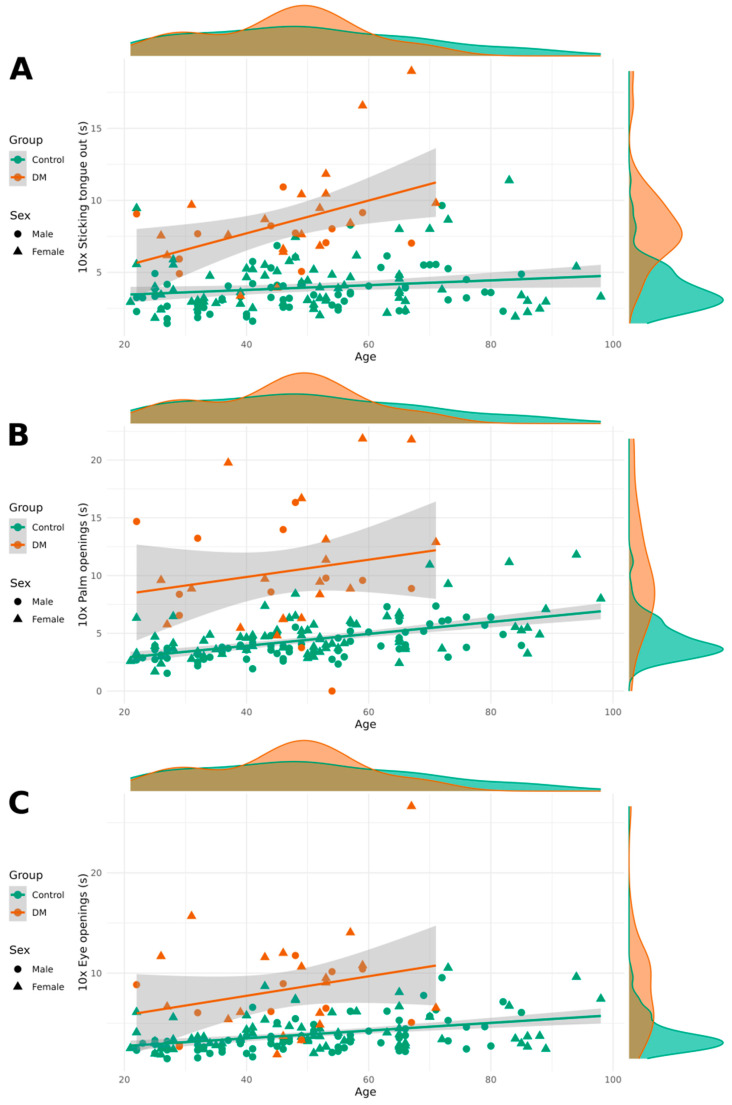
Completion times for the tongue-sticking tests (**A**), palm-opening tests (**B**), and eye-opening tests (**C**) categorized by patient and control groups as well as by sex.

**Figure 3 neurolint-17-00118-f003:**
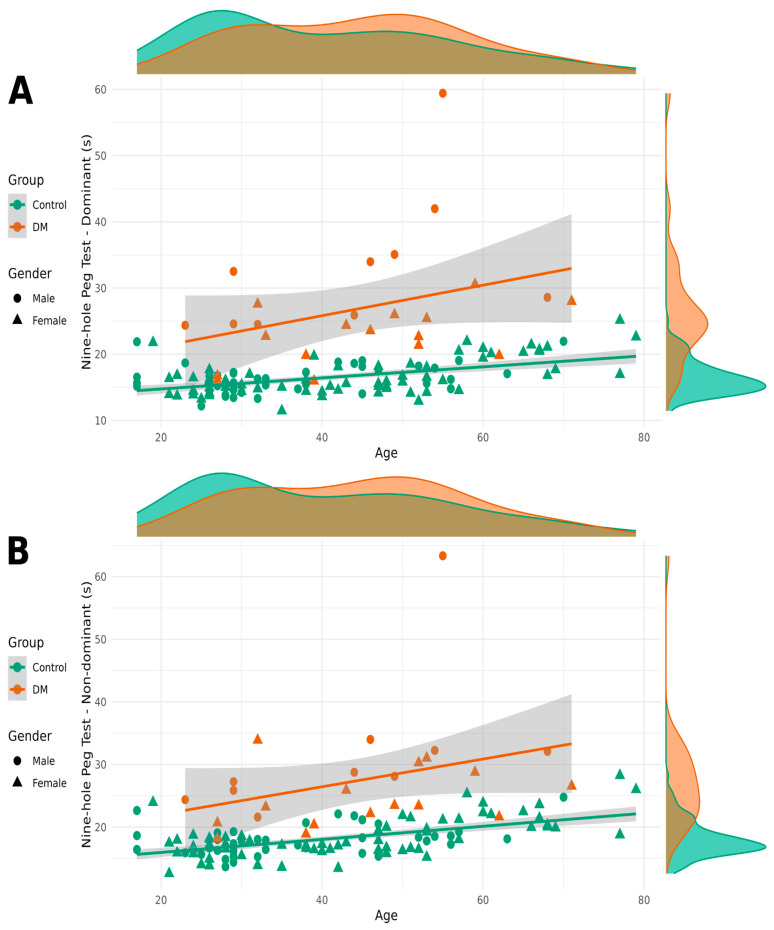
Results of the nine-hole peg test for the dominant hand (**A**) and non-dominant hand (**B**) categorized by patient and control groups as well as by sex.

**Figure 4 neurolint-17-00118-f004:**
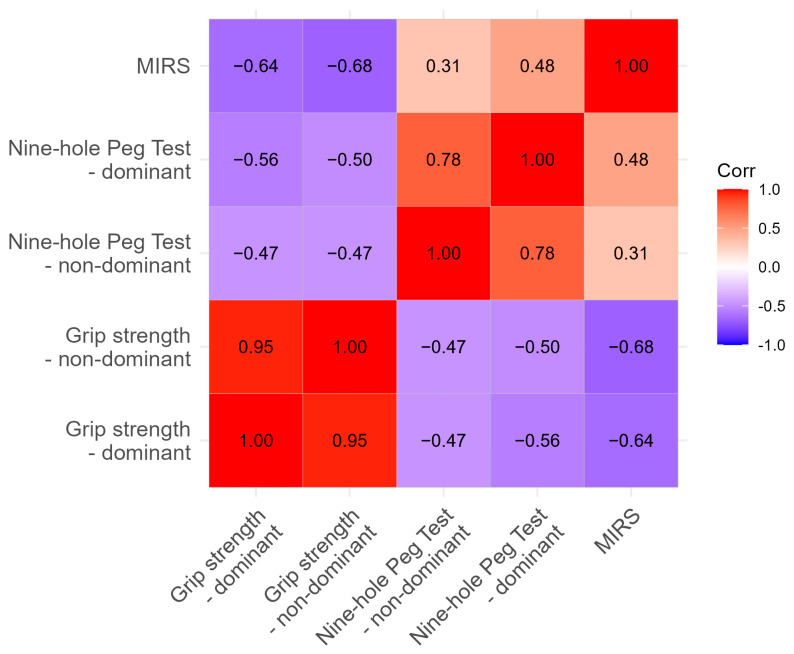
Correlations of NHPT with clinical data. Table shows correlation coefficients (r, Pearson’s test). DH: dominant hand, NDH: non-dominant hand, MIRS: Muscular Impairment Rating Scale, NHPT: nine-hole peg test.

**Table 1 neurolint-17-00118-t001:** Basic characteristics of participants. Age is shown in years.

Group	Sex	N	Mean	SD	95% CI	Min	Max	Shapiro–Wilk P
**Control**	**Female**	71	51.24	19.54	[46.69, 55.79]	21	98	0.012
	**Male**	68	49.6	17.25	[45.5, 53.7]	22	85	0.045
**Patient**	**Female**	19	47.47	12.08	[42.04, 52.91]	26	71	0.803
	**Male**	12	44.33	13.67	[36.6, 52.07]	22	67	0.721

**Table 2 neurolint-17-00118-t002:** Results of measurements. The results are presented as average ±SD (handgrip strength in kilograms (kg); all other recordings were measured in seconds). Also added are min–max, and CI (95%) results. DH: dominant hand, NDH: non-dominant hand, MIRS: Muscular Impairment Rating Scale, NHPT: nine-hole peg test.

Grip Strength DH (kg)								
**Group**	**Sex**	**N**	**Mean**	**SD**	**95% CI**	**Min**	**Max**	**Shapiro–Wilk P**
**Control**	**Female**	71	25.38	8.14	[23.48, 27.27]	3.8	41.3	0.002
	**Male**	68	43.81	9.74	[41.5, 46.13]	24.3	72.7	0.268
**Patient**	**Female**	19	10.39	6.46	[7.48, 13.29]	2.1	26.1	0.167
	**Male**	12	8.22	6.7	[4.43, 12.01]	0	21.7	0.492
Grip Strength NDH (kg)								
**Group**	**Sex**	**N**	**Mean**	**SD**	**95% CI**	**Min**	**Max**	**Shapiro–Wilk P**
**Control**	**Female**	71	23.21	7.72	[21.41, 25]	4.6	38.7	0.038
	**Male**	68	40.96	8.77	[38.88, 43.05]	22.4	62	0.371
**Patient**	**Female**	19	9.12	5.67	[6.57, 11.67]	2.9	21.4	0.022
	**Male**	12	8.8	6.3	[5.24, 12.36]	0	17.3	0.133
Eye Opening (s)								
**Group**	**Sex**	**N**	**Mean**	**SD**	**95% CI**	**Min**	**Max**	**Shapiro–Wilk P**
**Control**	**Female**	71	4.19	1.83	[3.76, 4.62]	1.88	10.53	<0.001
	**Male**	68	3.62	1.61	[3.23, 4]	1.48	9.55	<0.001
**Patient**	**Female**	19	9.27	5.67	[6.72, 11.82]	1.89	26.63	0.019
	**Male**	12	6.9	3.1	[5.15, 8.66]	2.69	11.76	0.448
Palm Opening (s)								
**Group**	**Sex**	**N**	**Mean**	**SD**	**95% CI**	**Min**	**Max**	**Shapiro–Wilk P**
**Control**	**Female**	71	4.89	2.03	[4.42, 5.36]	1.68	11.8	<0.001
	**Male**	68	4	1.28	[3.7, 4.3]	1.54	7.36	0.011
**Patient**	**Female**	19	10.89	5.46	[8.44, 13.35]	4.8	21.85	0.011
	**Male**	12	9.48	4.69	[6.82, 12.13]	0	16.33	0.741
Tongue Extensions (s)								
**Group**	**Sex**	**N**	**Mean**	**SD**	**95% CI**	**Min**	**Max**	**Shapiro–Wilk P**
**Control**	**Female**	71	4.17	1.84	[3.74, 4.6]	1.83	11.39	<0.001
	**Male**	68	3.72	1.53	[3.36, 4.09]	1.45	9.64	<0.001
**Patient**	**Female**	19	8.97	3.8	[7.26, 10.67]	3.36	18.98	0.037
	**Male**	12	7.57	1.74	[6.58, 8.55]	4.91	10.93	0.862
MIRS Score								
**Group**	**Sex**	**N**	**Mean**	**SD**	**95% CI**	**Min**	**Max**	**Shapiro–Wilk P**
**Patient**	**Female**	19	3.74	0.93	[3.32, 4.16]	2	5	0.024
	**Male**	12	3.58	1	[3.02, 4.15]	2	5	0.137
Nine-Hole Peg Test, right (s)								
**Group**	**Sex**	**N**	**Mean**	**SD**	**95% CI**	**Min**	**Max**	**Shapiro–Wilk P**
**Control**	**Female**	67	16.56	2.78	[15.89, 17.22]	11.44	25.13	<0.001
	**Male**	43	16.31	2.1	[15.69, 16.94]	12.18	21.95	0.189
**Patient**	**Female**	15	22.69	4.47	[20.43, 24.95]	15.95	30.56	0.749
	**Male**	10	33.09	10.91	[26.32, 39.85]	24.36	59.42	0.014
Nine-Hole Peg Test, left (s)								
**Group**	**Sex**	**N**	**Mean**	**SD**	**95% CI**	**Min**	**Max**	**Shapiro–Wilk P**
**Control**	**Female**	67	18.3	3.15	[17.54, 19.05]	12.61	28.25	0.005
	**Male**	43	17.79	2.33	[17.09, 18.49]	13.7	24.78	0.038
**Patient**	**Female**	15	24.55	4.71	[22.17, 26.93]	18.1	33.88	0.478
	**Male**	10	31.77	11.72	[24.5, 39.03]	21.6	63.33	<0.001

**Table 3 neurolint-17-00118-t003:** Linear regression models for grip strength, relaxation tests, nine-hole peg test (NHPT), and the Muscular Impairment Rating Scale (MIRS). The values represent regression coefficients, with standard errors provided in parentheses. Every group (age, sex, heathy patient) has CI below the group’s results. DH: dominant hand, NDH: non-dominant hand. Appendix A report confidence intervals, t-values, standard errors, and *p*-values.

Regression Results								
	*Dependent variable:*							
	Grip Strength, Right (kg)	Grip Strength, Left (kg)	Eye Opening (s)	Palm Opening (s)	Tongue Sticking (s)	Nine-Hole Peg Test, Right (s)	Nine-Hole Peg Test, Left (s)	MIRS Score
	(1)	(2)	(3)	(4)	(5)	(6)	(7)	(8)
Age	−0.287 *	−0.259 *	0.043 *	0.052 *	0.025 *	0.121 *	0.132 *	0.034 *
	(−0.356, −0.218)	(−0.322, −0.195)	(0.021, 0.064)	(0.031, 0.074)	(0.007, 0.042)	(0.074, 0.167)	(0.086, 0.179)	(0.009, 0.059)
Group, DM	−24.590 *	−22.541 *	4.539 *	6.019 *	4.519 *	9.886 *	8.798 *	
	(−27.723, −21.456)	(−25.411, −19.670)	(3.566, 5.511)	(5.039, 7.000)	(3.736, 5.301)	(8.109, 11.663)	(7.007, 10.589)	
Sex, Female	−14.289 *	−14.109 *	0.803 *	0.884 *	0.567 *	−2.506 *	−1.772 *	0.046
	(−16.705, −11.873)	(−16.323, −11.896)	(0.053, 1.553)	(0.128, 1.640)	(−0.037, 1.170)	(−3.943, −1.069)	(−3.220, −0.324)	(−0.586, 0.679)
Constant	56.182 *	52.146 *	1.352 *	1.366 *	2.415 *	13.098 *	13.816 *	2.069 *
	(52.315, 60.049)	(48.603, 55.688)	(0.151, 2.552)	(0.156, 2.576)	(1.449, 3.381)	(11.047, 15.150)	(11.748, 15.884)	(0.862, 3.277)
Observations	170	170	170	170	170	135	135	31
R^2^	0.734	0.743	0.380	0.506	0.456	0.557	0.513	0.211
Adjusted R^2^	0.729	0.738	0.369	0.497	0.447	0.546	0.502	0.154
F Statistic	152.345 * (df = 3; 166)	160.046 * (df = 3; 166)	33.952 * (df = 3; 166)	56.610 * (df = 3; 166)	46.448 * (df = 3; 166)	54.814 * (df = 3; 131)	45.971 * (df = 3; 131)	3.736 * (df = 2; 28)

Note: * = *p* < 0.05.

## Data Availability

Raw data supporting the conclusions of this study will be made available by the authors upon request.

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
