# Peer review of "Measuring Myotonia: Normative Values and Comparison with Myotonic Dystrophy Type 1"

_2035-8377, 2025, doi:10.3390/neurolint17080118_

Round 1

Reviewer 1 Report

Comments and Suggestions for Authors

The manuscript presents a study, which aimed to determine normal values for relaxation tests in healthy individuals and compare them to the values of myotonic dystrophy type 1 patients in order to evaluate the influence of disease-specific variations like muscle weakness and disability score. The manuscript is clear and concise.

However, I have a few comments for consideration.

  • I advise in ‘a Hungarian population’ rather than in ‘the Hungarian population’, if not all Hungarian DM1 patients are tested, rather a region’s. Otherwise please state.
  • Methods part should be structured with titles (similar to “Statistical Analysis”)
  • Limitations are found in the discussion by the specific parts, but it would be more elegant to collect them also in a short paragraph.

Author Response

Dear Reviewer,

Thank you very much for taking the time to review this manuscript. We have made the corrections as you suggested. Not all Hungarian DM1 patients were tested, thank you for your comment. We have made the methods part more structured with titles by the exact methods. We have also collected the limitations in the limitations part:

"This study has several limitations. As a single-center investigation, the generalizability of the findings may be limited. Additionally, test-retest reliability was not assessed, and in the healthy population, a potential learning effect could have influenced performance, possibly leading to an overestimation of functional capacity. The absence of CTG repeat expansion data restricts genotype-phenotype correlation, which could provide deeper insights into disease severity and progression. These factors should be considered when interpreting the results and planning future studies."

Sincerely, 

Andrea Sipos

Reviewer 2 Report

Comments and Suggestions for Authors

Dear Editor and authors,

Thank you for the opportunity to review the manuscript titled “Measuring Myotonia: Normative Values and Comparison to Myotonic Dystrophy Type 1.” The study addresses a relevant clinical topic. However, I have identified several areas requiring revision, including clarifications in the methods section, enhanced statistical reporting (e.g., confidence intervals and effect sizes), improved clarity in language use, and acknowledgment of key limitations.

Please find detailed comments, organized by page.

Page 1, Lines 2–5: Consider adding geographic context, such as “in a Hungarian cohort”, to improve clarity.

Page 1, Lines 10–24: Include more specific numerical results (e.g., effect sizes or exact means) to strengthen the abstract.

Page 1, Line 47: Elaborate on the limitations of previously used measurement methods to better justify the methodological approach.

Page 1, Lines 88–92: Clarify whether the functional and cognitive tests employed were validated for the Hungarian population.

Page 1, Line 113: The background section is well-supported with relevant references.

Page 1, Lines 114–118: Consider rephrasing the objective more concisely.

Page 2, Lines 124–125: The absence of exact CTG repeat numbers in DM1 patients should be addressed more explicitly as a limitation.

Page 2, Line 128: Provide information on the number of excluded participants and reasons for exclusion.

Page 3, Line 157: Specify the statistical software used and report the alpha threshold (e.g., p < 0.05) for reproducibility.

Page 4, Lines 166–169: Table 1 should include confidence intervals for descriptive statistics (e.g., age).

Page 4, Line 172: Replace commas with decimal points for consistency with English-language scientific formatting (e.g., 43.8 instead of 43,8).

Page 6, Table 3: Include confidence intervals for regression coefficients to better convey statistical precision.

Page 6, Lines 185–190: Consider reporting effect sizes (e.g., Cohen’s d) for group comparisons.

Page 7, Lines 196–200: Although log-normality is assumed, normality tests (e.g., Shapiro-Wilk, Q-Q plots) are not reported and should be added.

Page 9, Lines 214–217: The statement that this is the “first study” using this standardized approach should be supported with a reference.

Page 10, Line 259: The >60% difference in NHPT for males should be further discussed in terms of effect size and clinical relevance.

Page 10, Lines 260–262: Expand the discussion on clinical implications, especially regarding rehabilitation and diagnostics.

Limitations - Add a brief section with the following:

Absence of precise CTG expansion data in the DM1 group.

Use of a convenience sample from a single center, which may affect generalizability.

Possible learning effects in the control group, even with attempts to minimize them.

Lack of test-retest reliability assessment for the clinical tools used.

Supplementary Data

The supplementary table is useful but would benefit from graphical presentation (e.g., boxplots or regression lines).

Several handgrip strength values in older female DM1 patients are negative (e.g., –5.67 kg), which likely reflects a data entry or processing error and should be verified.

Other Considerations

Ethical approval is stated, but the ethics approval ID is missing.

There is no mention of funding sources, this should be added for transparency.

Best regards,

Author Response

Dear reviewer, 

Thank you very much for taking the time to review this manuscript.

Consider adding geographic context, such as “in a Hungarian cohort”, to improve clarity:

"Myotonia is a rare neuromuscular condition characterized by impaired muscle relaxation. In this study, we provide normative values for clinical tests related to myotonia in a Hungarian population and compare them to patients with myotonic dystrophy type 1 (DM1)."

Include more specific numerical results (e.g., effect sizes or exact means) to strengthen the abstract:

"We observed non-significant decline in handgrip strength and relaxation tests with age (p<0.05). Significant differences were found between controls (n:139) and DM1 patients (n=31) in all tests (p<0.05). Gender differences were noted in the healthy population: men (n:68/139) had stronger handgrip (mean of males 42,45 ±1.15 vs females 24,3 ±0.9) and slower relaxation tests (mean of eye openings in males 3.6 ±0.2 vs in females 4.2±0.2, tongue extensions in males 3.7±0.2 vs in females 4.2±0.2, palm openings in males 4±0.2 vs in fameless 4.9±0.2 However, these differences were not present among DM1 patients."

Elaborate on the limitations of previously used measurement methods to better justify the methodological approach: 

"The measurement methods that we are using are for the follow-up of the course of the disease. DM1  is a muscular dystrophy,  it is associated with muscle wasting, muscle weakness, muscle relaxation disorders (myotonia) and systemic symptoms. According to the literature, and also the everyday routine, we are using easily performed, practical tests.  Muscle strength can be measured manually or instrumentally, which is influenced by the examiner, the timing of the test, and the patient's cooperation. Myotonia testing can also be done using manual or instrumental measurements; with similar limiting factors. Other functional tests such as nine-hole peg test (NHPT) and the 6-minute walk test (6MWT) hvas other limitations, NHPT is mostly limited by the dexterity, which is combined by motorical and sensorical factors, also the cooperation of the pateint, the everyday use of ther hands. The 6MWT is mostly affected by cardiopulmonary status, and also the muscle strength of the lower extremities. These tests we are using, all of them has limiting factors, we haven't changed these tests, we are focusing on to improve in these tests, and the avoid or minimize the limiting factors as much as we can."

Clarify whether the functional and cognitive tests employed were validated for the Hungarian population:

"The background of the functional tests we are used are listed in the references (according to the international literature). There is no information about the validation in Hungarian population.

The cognitive tests are validated in Hungarian population. 

Márta, Volosin, Janacsek Karolina, and Németh Dezsô. "A Montreal Kognitív Felmérés (MoCA) magyar nyelvű adaptálása egészséges, enyhe kognitív zavarban és demenciában szenvedő idős személyek körében." Psychiatria Hungarica 28.4 (2013): 370-392.

Beáta, Kaszás, and Fekete Judit. "Validation of the Hungarian version of Addenbrooke's cognitive examination for detecting major and mild neurocognitive disorders." (2020)."

The background section is well-supported with relevant references.

Consider rephrasing the objective more concisely:

"The aim of our study is to determine normal values for relaxation tests and to assess the factors influencing these tests (age, gender). We compared the results of DM1 patients to those of healthy subjects to evaluate the influence of disease-specific variations."

The absence of exact CTG repeat numbers in DM1 patients should be addressed more explicitly as a limitation:

"We agree that this is a severe limitation, we mentioned in the article, and also involved it in the limitation section. Unfortunately the local genetic laboratory is not able to perform this kind of examination, but we are planning to do it in other genetic laboratory.

The absence of CTG repeat expansion data restricts genotype-phenotype correlation, which could provide deeper insights into disease severity and progression. These factors should be considered when interpreting the results and planning future studies."

Provide information on the number of excluded participants and reasons for exclusion:

"There were no exclusion, because we have only involved healthy individuals who were optimal for the incusion criteria. Among our patients we could include all of them who have had there annual follow-up during that period of time, when we were collecting the healthy individuals."

Specify the statistical software used and report the alpha threshold (e.g., p < 0.05) for reproducibility:

"Thank you for raising our attention to this missing piece of information. We now state the software used in the manuscript (line xx): "We conducted the statistical analyses using R version 4.3.3 (R Core Team, 2024)." We now also state that "The significance level was set at α = 0.05" (line xx).   Reference: R Core Team. R: A language and environment for statistical computing. Version 4.3.3. Vienna: R Foundation for Statistical Computing; 2024. Available from: https://www.R-project.org/  

Include confidence intervals for regression coefficients to better convey statistical precision:

"As per your request, we now include confidence intervals for regression coefficients in Table 3. As we agree that statistical precision is of utmost importance, we now report the confidence intervals, t-values, standard errors, and p-values in new, separate Supplementary Tables, numbered from 2 to 9."  

Table 1 should include confidence intervals for descriptive statistics (e.g., age).

Thank you for your note, we have made minimal and maximal intervals according to the ages. The other parameters are not suitable for the confidence interval, therefore we thought that we do not want to add it because of the integritiy of the table. 

Replace commas with decimal points for consistency with English-language scientific formatting (e.g., 43.8 instead of 43,8). 

"Thank you for your note, we have replaced them."

Consider reporting effect sizes (e.g., Cohen’s d) for group comparisons:    

"Unfortunately we could not decide what kind of group comparioson were you thinking of, please let us know."

Although log-normality is assumed, normality tests (e.g., Shapiro-Wilk, Q-Q plots) are not reported and should be added:

"We thank the reviewer for this important observation. We acknowledge that there was an error in our original manuscript regarding the use of log-transformation. Upon review, we clarify that while log-normalization of NHPT performance data was initially considered during our analytical planning, we ultimately decided against implementing this transformation in our final analyses. The manuscript has been corrected to accurately reflect that linear regression models were applied to untransformed data. In response to the reviewer's suggestion, we have now included the results of Shapiro-Wilk normality tests for each dependent variable across all study groups [Table X/Supplementary Table X]. These results support the appropriateness of our modeling approach and provide transparency regarding the distributional assumptions underlying our analyses. We apologize for any confusion caused by this error and appreciate the reviewer's careful attention to this methodological detail.

The statement that this is the “first study” using this standardized approach should be supported with a reference.

Thank you for your note. As we know there is no study about the normal values of functional tests, you can find in the references the data about these values; but we added "in a Hungarian population".

The >60% difference in NHPT for males should be further discussed in terms of effect size and clinical relevance:

"It is important to note that the healthy female population had a higher number of older individuals than males, so it is possible that the healthy male mean is less than the true mean, while the female mean is more than the true mean. This may partly explain such a difference; or, with regard to muscle strength in DM1 males, it should be noted that we would expect more pronounced muscle weakness with age and disease progression, compared to females. The significance of this difference is therefore not yet fully understood and further investigation is needed."

Expand the discussion on clinical implications, especially regarding rehabilitation and diagnostics:

"In summary, we established a standardized battery of tests to assess myotonia and distal muscle function, providing valuable tools for diagnosis, monitoring, and rehabilitation. These tests, supported by normative data, help detect early functional decline and guide individualized therapy, particularly in the context of age and sex differences. They also offer sensitive outcome measures for evaluating treatment efficacy in clinical trials, including emerging disease-modifying and gene-based therapies. Overall, this approach supports more precise and personalized care for patients with myotonic disorders."

Limitations:

"This study has several limitations. As a single-center investigation, the generalizability of the findings may be limited. Additionally, test-retest reliability was not assessed, and in the healthy population, a potential learning effect could have influenced performance, possibly leading to an overestimation of functional capacity. The absence of CTG repeat expansion data restricts genotype-phenotype correlation, which could provide deeper insights into disease severity and progression. These factors should be considered when interpreting the results and planning future studies."

The supplementary table is useful but would benefit from graphical presentation (e.g., boxplots or regression lines). Several handgrip strength values in older female DM1 patients are negative (e.g., –5.67 kg), which likely reflects a data entry or processing error and should be verified.

"We thank the reviewer for this feedback and recognize that the purpose of Supplementary Table 1 requires clarification. This table does not contain observed data, but rather model-predicted values for hypothetical individuals across all combinations of age, gender, and disease status, as generated by our fitted linear regression models. The table serves as a clinical reference tool to provide normative values for comparison purposes, as now we clarify in our Results section:

Using the fitted regression models, we generated predicted motor test scores for hypothetical individuals across all combinations of the predictor variables (e.g., 20-year-old healthy male, 20-year-old female SM1 patient, 30-year-old male SM1 patient). These predictions were compiled into a comprehensive reference table to serve as a clinical decision-support tool, enabling clinicians to rapidly assess patient performance relative to age- and gender-matched healthy controls and to established norms for comparable SM1 patients."

We now also reflect on the utility of the Supplementary table in the Discussion:

"Supplementary Table 1 provides clinicians with readily accessible predictive normative reference values that account for the expected effects of age, gender, and disease status on motor performance, facilitating more accurate interpretation of individual patient results within the context of relevant comparison groups. The reference table enables immediate comparison of a patient's observed performance against both healthy peers and individuals with similar demographic characteristics and disease status."

Regarding the negative handgrip strength values in older female DM1 patients, these reflect the mathematical output of linear regression extrapolation rather than data entry errors. When the regression model predicts performance for individuals with combinations of advanced age, female gender, and DM1 status—factors that all independently contribute to reduced handgrip strength—the linear model can produce negative values that are biologically impossible.

To address this limitation while maintaining clinical utility, we have modified the table to set negative predicted values to 0 kg, with a footnote clearly stating: "Values originally predicted as negative by the linear model have been set to 0 kg, as negative handgrip strength is not biologically possible. This reflects the limitations of linear extrapolation in extreme demographic/clinical combinations."

Regarding the suggestion for graphical presentation, we respectfully note that the regression models and their relationships are already comprehensively illustrated in the main figures of the manuscript. The tabular format of Supplementary Table 1 is specifically designed to serve as a practical clinical reference tool, allowing clinicians to quickly locate specific predicted values for patients with particular demographic and clinical characteristics.

Ethical approval is stated, but the ethics approval ID is missing.

Ethical ID is added, the ID number: 53593-5/2020/EÜIG

There is no mention of funding sources, this should be added for transparency.

This study had no foundation.

Reviewer 3 Report

Comments and Suggestions for Authors

This manuscript presents a valuable, well-structured study comparing normative values of functional myotonia tests with data from DM1 patients. The methodology is generally sound, the statistical analysis is appropriate, and the topic has clear clinical relevance. The authors provide a practical, standardized battery of tests applicable in clinical settings.

Importantly, the study uses accessible clinical tests (eye opening, tongue extension, palm opening), which can be widely adopted. The sample size, including 139 controls and 31 patients,  is large enough to allow for robust statistical comparisons. The use of multiple regression models adds robustness to the results and allows exploration of age, sex, and disease effects. The findings support the utility of these tests in DM1 diagnosis and monitoring, offering potential normative benchmarks, possibly useful also for treatment assessments.

In the conclusion, I would add which test is more useful to measure myotonia in DM1, according to the Authors

Author Response

Dear reviewer, 

Thank you very much for taking the time to review this manuscript. Among the relaxation test, there is no significant difference, but we have unpublished results from the patients's follow-ups, and according to that the tongue extension seems to be the most sensitive examination, because it showed singificant differences and also (theoratically) the tongue muscles can be the less affected by the warming up phenomenon. The nine-hole peg test is more compleceted (it is for the examination of hand dexterity, and mytonia is just a part of the influencing factors) examination in the view of myotonia, so therefore it is the less sensitive for the examination of myotonia. Therefore we do not want to emphasize which test should be performed. 

Sincerely, 

Andrea Sipos